# Major antigenic site B of human influenza H3N2 viruses has an evolving local fitness landscape

Nicholas C. Wu [1], Jakub Otwinowski[2], Andrew J. Thompson [3], Corwin M. Nycholat[3],
Armita Nourmohammad[2,4] & Ian A. Wilson [1,5 ✉]

Antigenic drift of influenza virus hemagglutinin (HA) is enabled by facile evolvability. However, HA antigenic site B, which has become immunodominant in recent human H3N2 influenza viruses, is also evolutionarily constrained by its involvement in receptor binding. Here, we employ deep mutational scanning to probe the local fitness landscape of HA antigenic site B in six different human H3N2 strains spanning from 1968 to 2016. We observe that the fitness landscape of HA antigenic site B can be very different between strains. Sequence variants that exhibit high fitness in one strain can be deleterious in another, indicating that the evolutionary constraints of antigenic site B have changed over time. Structural analysis suggests that the local fitness landscape of antigenic site B can be reshaped by natural mutations via modulation of the receptor-binding mode. Overall, these findings elucidate how influenza virus continues to explore new antigenic space despite strong functional constraints.

[1] Department of Integrative Structural and Computational Biology, The Scripps Research Institute, La Jolla, CA 92037, USA. [2] Max Planck Institute for Dynamics and Self-Organization, 37077 Göttingen, Germany. [3] Department of Molecular Medicine, The Scripps Research Institute, La Jolla, CA 92037, USA. [4] Department of Physics, University of Washington, Seattle, WA 98195, USA. [5] The Skaggs Institute for Chemical Biology, The Scripps Research Institute, La Jolla, CA 92037, USA. ✉email: wilson@scripps.edu

H3N2 influenza virus was introduced into the human population in 1968. As the main antigen of influenza virus, the hemagglutinin (HA) from H3N2 viruses has five major antigenic sites (sites A–E)[1–3]. Throughout the past 50 years, the human H3N2 virus has acquired up to 9 new N-glycosylation sites due to constant pressure for immune evasion[4–7]. These N-glycans shield a large portion of the immunodominant HA1 globular head domain to limit antibody access to the major antigenic sites. Nonetheless, the added N-glycans can also negatively influence virus fitness when they are too close to the receptor-binding site (RBS)[8,9]. Consequently, there is a functional limit on the number of N-glycosylation sites present at any one time on the HA[7]. As N-glycosylation sites on the HA1 globular head domain accumulate over time and cover large regions of the HA1 surface, the more exposed membrane distal RBS in the HA head and the membrane proximal HA stem region may become increasingly prone to be targeted by antibodies[6,10]. Therefore, it is not surprising that antigenic site B, which significantly overlaps with the RBS, has become immunodominant in recent years[11,12]. While the evolution of antigenic site B is driven by immune evasion, the indispensable function of the RBS imposes a strong functional constraint on its evolution (i.e., limiting the number of tolerable mutations). For example, we have previously shown that such functional constraints can limit the mutational reversibility at residue 190, which is part of the antigenic site B as well as the RBS[13]. However, our understanding of the evolutionary constraints of antigenic site B is far from complete.

A fitness landscape refers to the genotype-to-phenotype map of many related genetic variants. Characterization of the fitness landscape of the influenza HA facilitates the understanding of its evolutionary constraints and predictability of naturally occurring mutations[14]. Deep mutational scanning, which combines saturation mutagenesis and next-generation sequencing so as to measure the phenotype of many genetic variants in parallel[15], has been applied to study the fitness landscape of influenza HA[13,14,16–18]. However, most deep mutational scanning studies on influenza HA have been focused on variants that are only one mutation away from the wild-type (WT) sequence[14,16,17]. As a result, epistatic effects (non-additive fitness effects) between mutations, which are believed to be pervasive, are often ignored[19]. In addition, it is unclear to what extent epistatic effects are localized in antigenic site B and whether any substantial interaction occurs with the rest of the influenza HA sequence, which also changes during the course of evolution.

In this study, we are interested in understanding how the local fitness landscape of antigenic site B in human H3N2 virus has changed over time due to epistasis when the genetic backgrounds differ for the rest of the protein. The replication fitness of 576 variants of interest in antigenic site B is measured by deep mutational scanning in each of the six different H3N2 genetic backgrounds, namely, A/Hong Kong/1/1968 (HK68), A/Bangkok/1/1979 (Bk79), A/Beijing/353/1989 (Bei89), A/Moscow/10/1999 (Mos99), A/Brisbane/10/2007 (Bris07), and A/North Dakota/26/2016 (NDako16). Our results indicate that the local fitness landscape of antigenic site B is highly dependent on the genetic background, which reveals the pervasiveness of evolutionary entrenchment and contingency during natural evolution of the human H3N2 virus. Specifically, sequence variants that are previously fit may progressively become unfit and become extinct (evolutionary entrenchment)[20], and sequence variants that are previously unfit may become fit and emerge (evolutionary contingency)[20]. Structural analysis further reveals that the receptor-binding mode of the human H3N2 virus has itself evolved over time, providing a mechanistic basis for the evolution of the local fitness landscape of antigenic site B. This study provides insights into how antigenic site B is still able to continue evolving in the presence of the strong functional constraints that enable virus entry.

## Results

**Deep mutational scanning of antigenic site B.** Several residues in antigenic site B of influenza HA are also part of the RBS, including residues 156, 158, 159, 190, 193, and 196. Previous studies have shown that mutations at these six residues can affect receptor binding and viral replication fitness[13,21–23]. We compiled an inventory of major amino acid variants that have reached high occurrence frequency (>90% in any given year) during the natural evolution of human H3N2 viruses over the past 50 years at these six residues of antigenic site B (Fig. 1a–c). This list includes four amino acid variants at residue 156 (Glu, Lys, Gln, His), four at residue 158 (Gly, Glu, Lys, Asn), three at residue 159 (Ser, Phe, Tyr), two at residue 190 (Asp and Glu), three at residue 193 (Ser, Asn, Phe), and two at residue 196 (Val and Ala). A total of $4 \times 4 \times 3 \times 2 \times 3 \times 2 = 576$ amino acid combinations (also known as haplotypes) are then possible across these six residues. These 576 variants were introduced into the HA of 6 different strains (genetic backgrounds) from 1968 to 2016, namely, A/Hong Kong/1/1968 (HK68), A/Bangkok/1/1979 (Bk79), A/Beijing/353/1989 (Bei89), A/Moscow/10/1999 (Mos99), A/Brisbane/10/2007 (Bris07), and A/North Dakota/26/2016 (NDako16), which were isolated approximately 10 years apart. HK68, Bk79, Bei89, Mos99, and Bris07 are all historical vaccine strains, whereas NDako16 is almost identical to A/Switzerland/9715293/2013 (Switz13), which was the vaccine strain for the 2015–2016 influenza season. The amino acid sequence of the HA ectodomain from NDako16 differs from Switz13 by only two mutations in HA1, namely, S198P and S312N. Out of these 576 variants, 38 of them have been observed in naturally circulating human H3N2 strains (Supplementary Fig. 1). While the WT sequences of HK68, Bk79, Bei89, Bris07, and NDako16 at the six residues of interest were included in our deep mutational scanning experiment, the WT sequence of Mos99 was not included because it contained a naturally rare variant T196.

Deep mutational scanning, which allows the phenotypes of many genetic variants to be measured in parallel[15], was performed to access the relative fitness of these 576 variants in different genetic backgrounds. Deep mutational scanning involves construction of a mutant library and measurement of the frequency change of individual mutants in the mutant library under a specified selection condition using next-generation sequencing. For each mutant, the magnitude of the frequency change is the proxy of its phenotype. Most deep mutational scanning studies, including those on influenza HA[14,16,17], have analyzed mutant libraries that mainly consisted of variants that are only one mutation away from the WT sequence. As a result, epistatic interactions could not be studied since mutants with two or more mutations were rare in these single mutant libraries. Here, using PCR primers that carry degenerate nucleotides, we were able to precisely introduce the desired set of amino acid mutations into the six residues of interest (see Methods).

Ultimately, we obtained six different local fitness landscapes, each with 576 variants. Two biological replicates were performed for each experiment with a high correlation (Pearson correlation = 0.92 to 0.97) observed between replicates (Supplementary Fig. 2). To compare the fitness effect of each combination of amino acids in different genetic backgrounds, we computed the variant preference, which describes the normalized fitness of individual variants (total of 576 variants) such that the average preference in a given genetic background equals 0 with a standard deviation of 1 (see Methods).

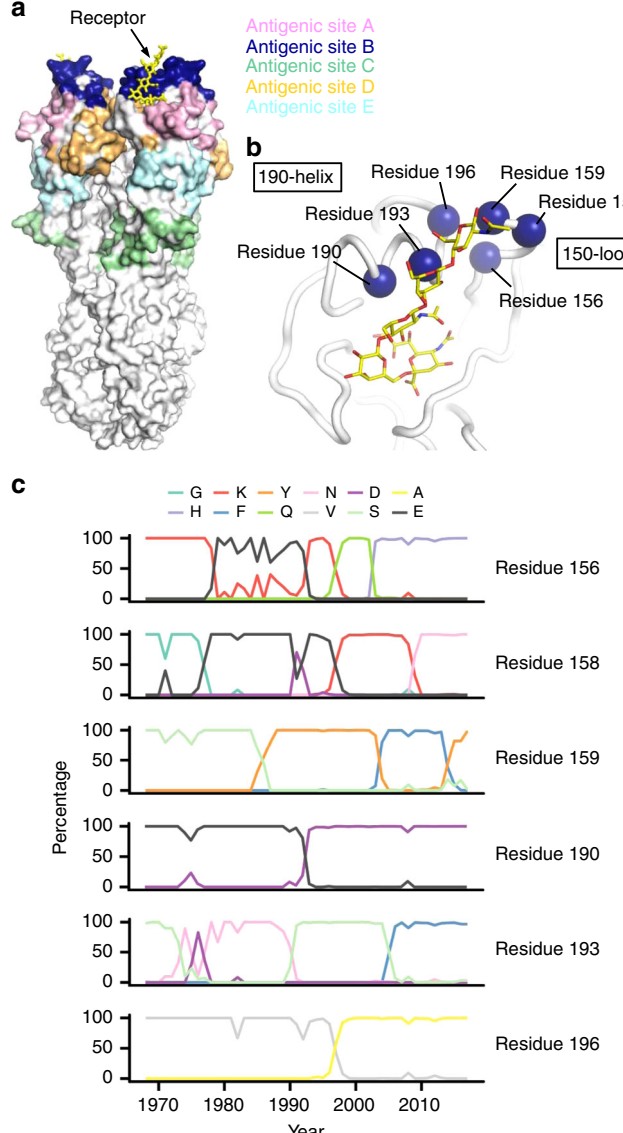

**Fig. 1 Natural evolution of antigenic site B in human H3N2 viruses. a** The location of the five major antigenic sites (sites A–E)[1–3] are shown on the HA trimer structure. The sialic acid receptor is shown in yellow in the receptor-binding site (RBS) that is proximal to and partially encompassing antigenic site B. **b** The RBS of HA is shown with the six residues of interest in antigenic site B highlighted in blue spheres and the receptor in yellow sticks. **c** The natural occurrence frequencies of the major amino acid variants, which are defined as having a natural occurrence at ≥50% in any given year, at the residues of interest in human H3N2 HA antigenic site B over time are shown.

**The local fitness landscape of antigenic site B is evolving**. We first aimed to compare the similarity of the local fitness landscapes among different genetic backgrounds. The local fitness landscape for each genetic background was summarized by a sequence logo (see Methods, Fig. 2a), which provided an overview of the fitness effect of each amino acid variant. Sequence logo analysis showed that several amino acid variants were favorable in certain genetic backgrounds but unfavorable in the others. For example, this analysis captures the known amino acid preference at residue 190, where Glu is favorable in early but not in recent human H3N2 strains, in terms of both viral replication fitness and receptor binding[13]. We have further utilized network

diagrams to visualize the evolution of the local fitness landscape (Supplementary Fig. 3). Each network diagram documents the preference of all 576 variants in a given genetic background. It is apparent from these network diagrams that variants with a high preference (fit) in one genetic background may have a low preference (unfit) in others. In addition, the distributions of the variant preference are different in different genetic backgrounds (Fig. 2b). For example, the distribution of variant preference in Bris07 is highly skewed as compared to HK68, suggesting that a relatively few variants are highly preferred in Bris07 (skewness = 1.05), but such a bias does not exist in HK68 (skewness = −0.13). We also examined the correlation between the local fitness landscapes from different genetic backgrounds (Fig. 2c). The Pearson correlations between the local fitness landscapes from different genetic backgrounds vary dramatically, ranging from −0.30 (Bk79 vs Bris07) to 0.92 (Bk79 vs Bei89). It is quite unexpected that a negative correlation could be observed. Low correlations between different local fitness landscapes indicate the prevalence of epistasis, which states that the fitness effect of a mutant depends on the amino acid sequence in other regions of the protein. Overall, epistasis between antigenic site B and the genetic background (i.e., the rest of the HA sequence) allows the local fitness landscape of antigenic site B to evolve over time as other residues of the HA protein evolve.

**Correlating local fitness landscape to natural evolution**. Next, we aimed to investigate how the compatibility of natural sequence variants at residues 156, 158, 159, 190, 193, and 196 varied in different focal genetic backgrounds (HK68, Bk79, Bei89, Mos99, Bris07, and NDako16). In other words, we were interested in addressing whether the sequence variants at residues 156, 158, 159, 190, 193, and 196 (i.e., haplotypes of these six residues of interest) from a naturally occurring strain that was isolated many years ago (up to five decades) would still be fit on the genetic background of a more recent strain, and vice versa. Subsequently, we plotted the preference of individual natural sequence variants (haplotypes of residues 156, 158, 159, 190, 193, and 196) in different focal genetic backgrounds (HK68, Bk79, Bei89, Mos99, Bris07, and NDako16) against the year of strain isolation. This analysis tracked how natural sequence variants from a given year would fit in a given focal genetic background from a different year. A schematic diagram of this analysis is shown in Supplementary Fig. 4. The preferences of natural sequence variants were mostly positive around the year that a given focal genetic background (HK68, Bk79, Bei89, Mos99, Bris07, or NDako16) was first isolated (Fig. 3a). For example, we observed that natural sequence variants from 1979 to 1991 had a positive preference in the genetic background of Bk79. In contrast, natural sequence variants that were isolated before 1979 or after 1991 had a negative preference in the genetic background of Bk79. Similar observations can be made for other focal genetic backgrounds. This analysis demonstrates that a given genetic background is generally compatible with the natural sequence variants of antigenic site B that were observed in a similar time period but not with those that were isolated long before or after or even immediately before (e.g., genetic background Bk79) or just after (e.g., genetic background Bei89).

We further aimed to quantify how the preferences of the natural sequence variants (i.e., haplotypes) at residues 156, 158, 159, 190, 193, and 196 in a given focal genetic background change as the strains diverge from the focal genetic background. The divergences of naturally occurring strains from a given focal year were computed based on the residue-averaged Kullback–Leibler (KL) divergence (see Methods). The entire HA amino acid sequences except residues 156, 158, 159, 190,

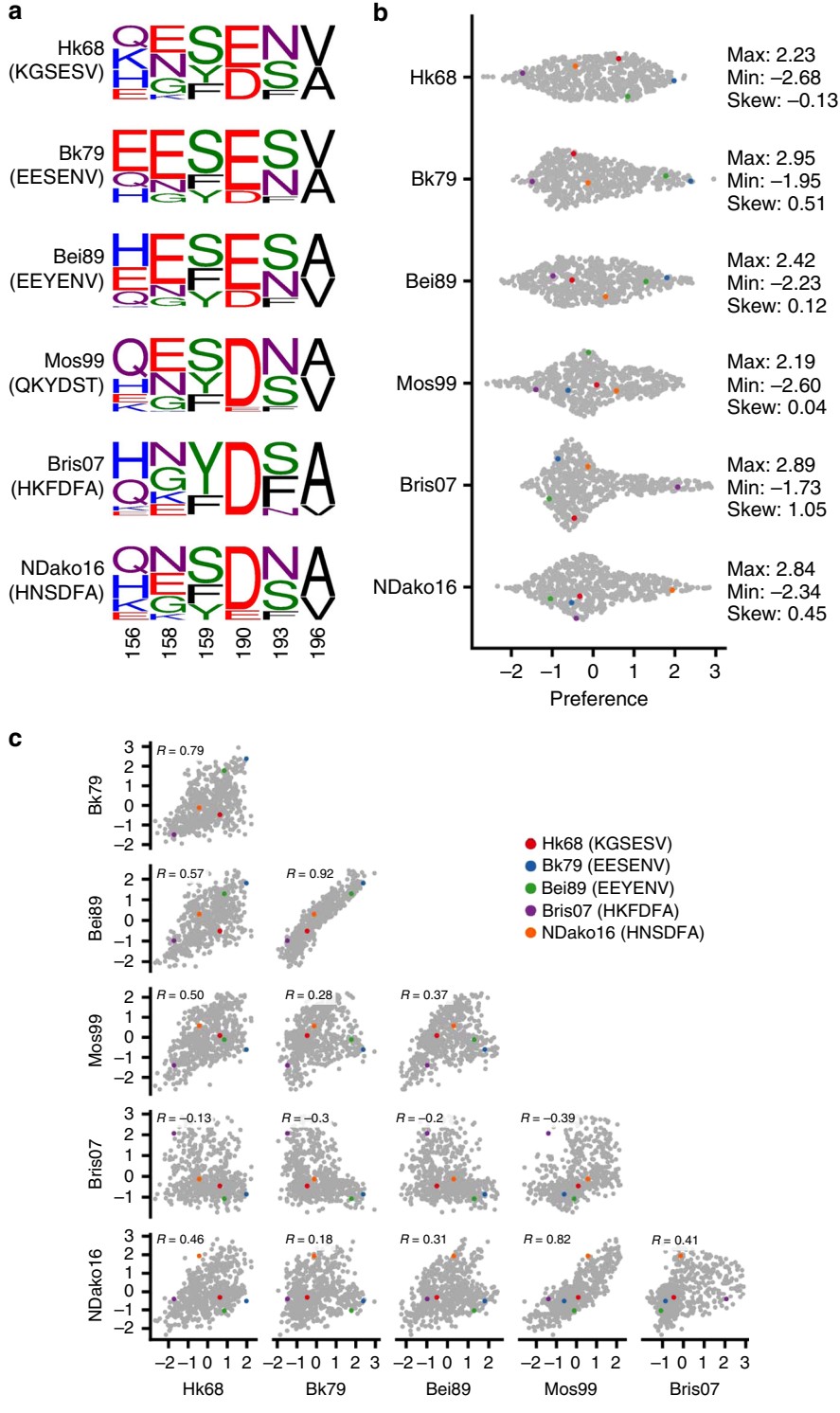

**Fig. 2 Comparing the local fitness landscapes of antigenic site B in six human H3N2 strains. a** The fitness effects of 576 variants, which include all combinations of major, naturally occurring amino acids at residues 156, 158, 159, 190, 193, and 196, are probed by deep mutational scanning. Sequence logo represents the amino acid preferences at each site (from left to right: residues 156, 158, 159, 190, 193, 196). **b** The distributions of preference for individual variants in different genetic backgrounds are shown as a sina plot, which allows normalized density of points to restrict the jitter along the y-axis. Each data point represents one variant. A total of 576 data points are present for each row (each genetic background). The maximum value (max), minimum value (min), and skewness (skew) for each distribution are indicated on the right. Negative skewness indicates that the mean is less than the median, whereas positive skewness indicates that the mean is larger than the median. Variant preference is the normalized fitness with a mean of 0 and a standard deviation of 1. **c** Correlations between the 576 variant fitness landscapes of different genetic backgrounds are shown, with each data point representing one variant. Pearson correlation coefficients (R) were computed using the log preferences for the 576 variants. **b**, **c** Data points corresponding to the WT sequences of HK68, Bk79, Bei89, Bris07, and NDako16 are colored as indicated. Of note, the WT sequence of Mos99 contains a naturally rare variant T196. Therefore, the WT sequence of Mos99 was not included in our deep mutational scanning experiment.

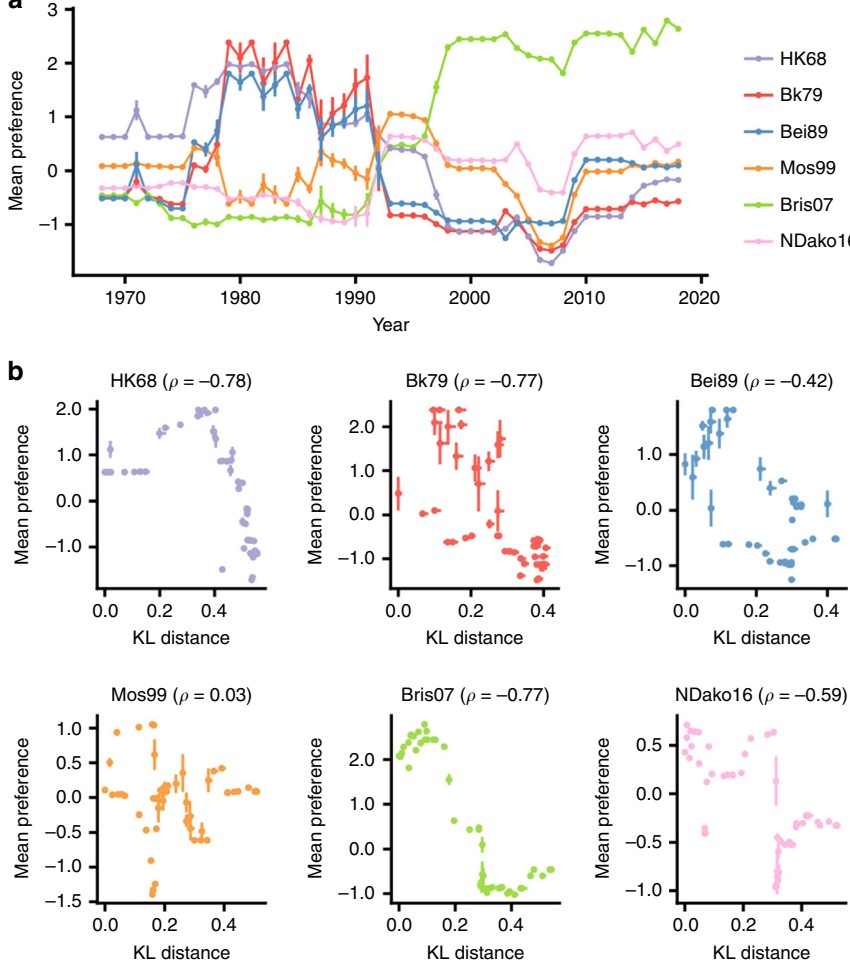

**Fig. 3 The local fitness landscape of antigenic site B coevolves with the genetic background. a** The variant preference for each naturally occurring variant was computed based on different genetic backgrounds. The average variant preference for different strains in a given year is shown. Different genetic backgrounds are represented by different lines, which are color coded as indicated on the right. Error bars represent the standard error of mean. A total of 45,218 human H3N2 strains isolated from 1968 to 2018 were analyzed. A schematic representation of this analysis is shown in Supplementary Fig. 4. **b** The divergence of sequence variants in natural strains from the focal year in each panel is calculated by the residue-averaged Kullback–Leibler (KL) divergence (see Methods). The entire HA sequence except residues 156, 158, 159, 190, 193, and 196 was used for calculating KL divergence. The relationship between the sequence divergence and the variant preference on the given genetic background is shown. Spearman rank correlation ($\rho$) between sequence divergence and the variant preference for each for each of the focal genetic background is shown in parentheses. The vertical error bar represents the standard error of the mean of the variant preference. The horizontal error bar represents the 95% confidence intervals for the estimated KL divergence based on a parametric bootstrap. In most cases, the 95% confidence intervals are quite tight. Source data are provided as a Source Data file.

193, and 196 were used to compute the KL divergence. A larger KL divergence indicates lower HA sequence identities between the natural variants and the focal genetic background. In most cases, the preferences of the natural sequence variants negatively correlate with KL divergence (Fig. 3b). Thus this analysis quantitatively shows that a given natural haplotype at residues 156, 158, 159, 190, 193, and 196 would generally become unfit as the genetic background diverges. Furthermore, such a decrease in fitness correlates with the degree of divergence of genetic background. Nonetheless, Mos99 seems to be an exception. Natural sequence variants from HA with high KL divergence from Mos99 have intermediate fitness in the Mos99 genetic background. It is known that epistasis promotes the diversity of functional sequence space[24,25], which may explain why the natural sequence variants from HA with high KL divergence from Mos99 are not particularly unfit. Overall, our results suggest that evolutionary entrenchment and contingency are pervasive in the natural evolution of H3N2 HA antigenic site B (Supplementary Fig. 5).

**Evolution of additive fitness and pairwise epistatic effects.** Fitness landscapes of molecules are often complex with both specific and non-specific interactions[26]. However, interpreting the structure of these high-dimensional landscapes is difficult[27] and highly sensitive to experimental noise. Previous work has shown that approximate fitness landscapes with additive and pairwise interactions among residues can accurately describe biophysical properties of proteins, including protein residue–residue contacts[28,29]. Additive fitness effects describe the independent contributions of each amino acid variant to fitness, whereas pairwise epistatic interaction effect describes how pairs of amino acids synergistically (positively) or antagonistically (negatively) impact the fitness[30,31]. Here we construct a statistical model to infer the additive fitness effects and pairwise epistatic interactions that underlie the six local fitness landscapes (see Methods). This model provides an interpretable description of the high-dimensional experimental landscape in the presence of the fitness measurement noise. Based on the 95% confidence, we classified each parameter into significantly

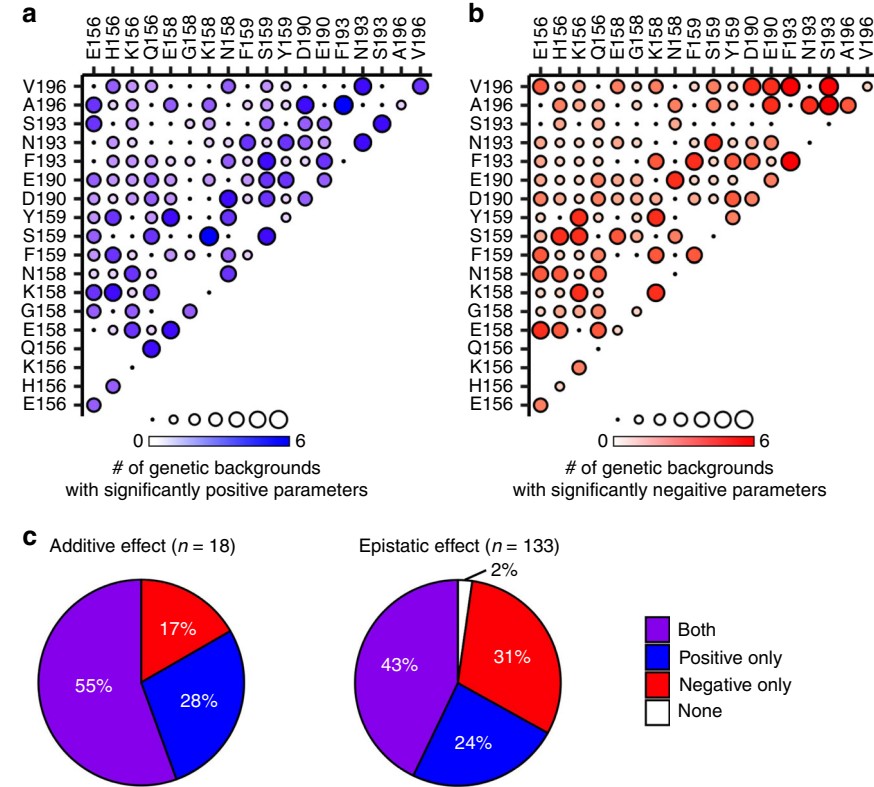

**Fig. 4 Inference of additive and pairwise epistatic effect.** Additive effect for each single amino acid variant and pairwise epistatic effect for each double amino acid variant are inferred based on the deep mutational scanning data. **a, b** For each additive effect and pairwise epistatic effect, the number of genetic backgrounds (out of six, Supplementary Fig. 6) that have an **a** significantly positive parameter or **b** negative parameter at 95% confidence interval are shown. The identity of a given double amino acid variant is represented by the labels on the x- and y-axes. For example, double amino acid variant E156/V196 is represented by the circle at the top left corner of the graph. Additive effects, which describes fitness contribution from a single amino acid variant, are shown on the diagonal, where the x- and y-axis labels are the same. For example, single amino acid variant E156 is represented by the circle at the bottom left corner of the graph, where both the x- and y-axes are E156. No circle is shown where the x- and y-axis labels represent the same residue position but different amino acid variant (e.g., E156/H156), which cannot occur at the same time. **c** Each additive effect (n = 18) and pairwise epistatic effect (n = 133) is classified into four categories, namely, both, positive only, negative only, and none. Both indicates both significantly positive and significantly negative parameters can be observed among those six genetic backgrounds. Additive effect of D190 is an example of both (Supplementary Fig. 6). Positive only indicates only significantly positive parameters but not significantly negative parameters can be observed. Epistatic effect of F193/A196 is an example of positive only (Supplementary Fig. 6). Negative only indicates only significantly negative parameters, but not significantly positive parameters, can be observed. Additive effect of F193 is an example of negative only (Supplementary Fig. 6). None indicates neither significantly positive parameters nor significantly negative parameters can be observed. Source data are provided as a Source Data file.

positive, significantly negative, and insignificant deviation from zero (neutral). These results are shown in Supplementary Fig. 6 and summarized in Fig. 4a, b.

Interestingly, a significant number of parameters for additive fitness effects and also for pairwise epistatic effects had different signs in different genetic backgrounds (both in Fig. 4c). These parameters include 10 out of 18 (55%) for additive fitness effects and 57 out of 133 (43%) for pairwise epistatic effects. This analysis demonstrates that, among major naturally observed amino acid variants in antigenic site B, both additive fitness and pairwise epistatic effects have large fluctuations during the course of human H3N2 evolution to date. As a result, the local fitness landscape of antigenic site B has been evolving extensively. Nevertheless, we acknowledge that the additive fitness and pairwise epistatic effects are both likely to be more conserved when unobserved amino acid variants in antigenic site B are also considered in our analysis, since they might be deleterious regardless of the genetic backgrounds.

**Structural evolution of HA receptor-binding domain.** To elucidate the mechanism that underlies the evolution of the local fitness landscape of antigenic site B, we wanted to compare the

receptor-binding mode of the original pandemic HK68 HA with that of a more recent HA (Bris07). The crystal structure of Bris07 HA in complex with human receptor analog 6′SLNLN (NeuAcα2-6Galβ1-4GlcNAcβ1-3Galβ1-4GlcNAc) was determined in our previous study[32]. Thus we determined HK68 HA in complex with 6′SLNLN at 2.25 Å resolution (Supplementary Table 1 and Supplementary Fig. 7). By aligning their receptor-binding domains (HA1 residues 117–265)[33], differences in the 6′SLNLN conformations can be observed (Fig. 5a, b). GlcNAc-3, Gal-4, and GlcNAc-5 are closer to the 190-helix when 6′SLNLN binds to Bris07 HA. In addition, there is a 90° rotation of GlcNAc-5 when 6′SLNLN binds to Bris07. Such conformational differences of 6′SLNLN can be attributed to the differences in the base of the HK68 HA RBS vs that of Bris07 HA RBS. As compared to HK68 HA RBS, Bris07 HA RBS has acquired several natural mutations at its base. Many of these mutations introduce additional interactions with Sia-1 and Gal-2 of 6′SLNLN (Fig. 5c, d). For example, mutation G135T introduces a water-mediated hydrogen bond (H-bond) with Sia-1 O4 atom, whereas N137S introduces an H-bond with the Sia-1 carboxylate group. Moreover, mutations W222R and G225N enable H-bonds with Gal-2.

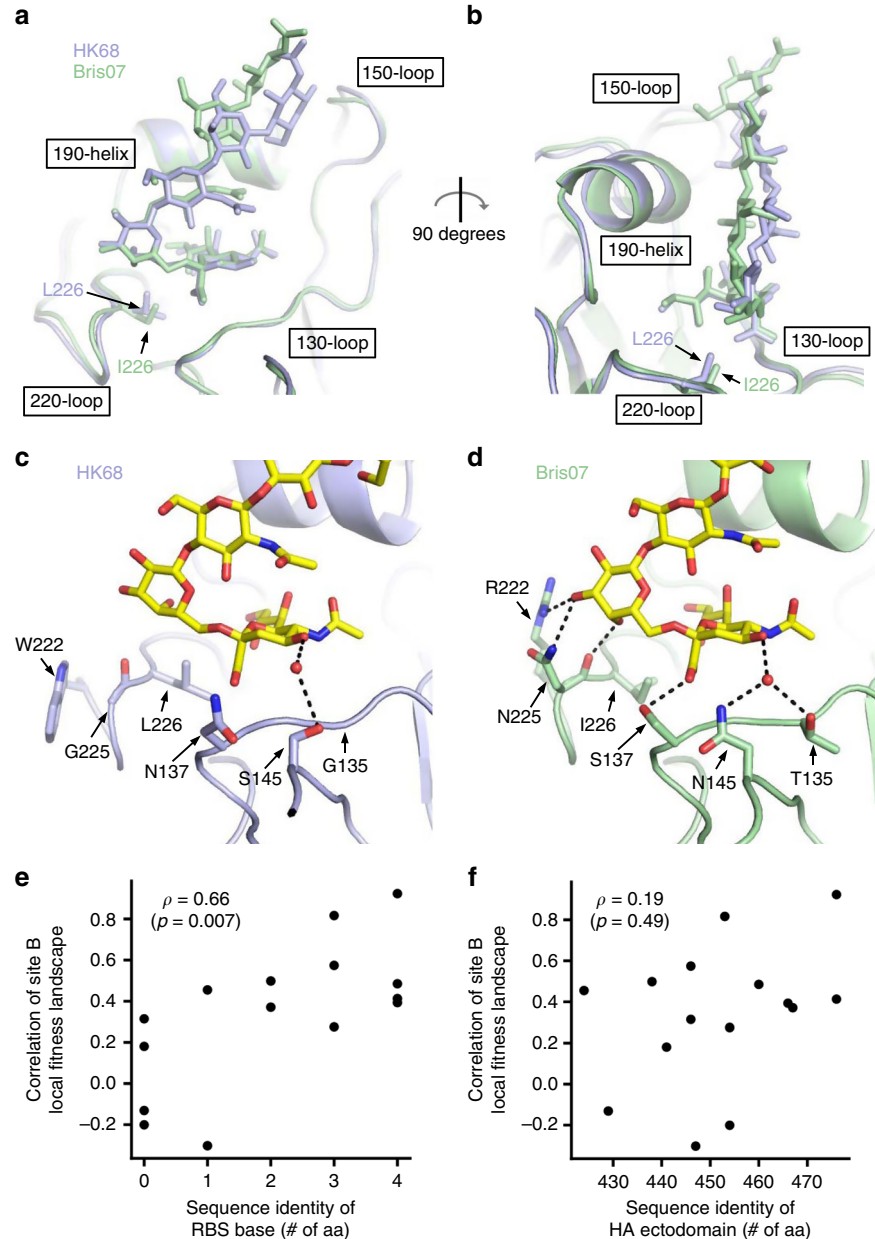

**Fig. 5 Structural evolution of influenza hemagglutinin receptor binding. a, b** HA structures of Bris07 and HK68 in complex with 6'SLNLN are aligned based on the receptor-binding subdomain (residues 117–265)[33]. **a** Front view of the HA RBS. **b** Side view of the HA RBS. HA is shown as cartoon representation. The side chain of HA1 residue 226 and 6'SLNLN are shown in stick representation. **c, d** Interactions between HA and 6'SLNLN at the base of **c** HK68 RBS or **d** Bris07 RBS are shown. Hydrogen bonds are represented by black dashed lines. Water molecules are shown as red spheres. 6'SLNLN is colored in yellow and shown in stick representation. Side chains of interest are also shown in stick representation. The structure of Bris07 in complex with 6'SLNLN is from PDB 6AOV[32]. **e, f** The relationship between the pairwise correlation of antigenic site B local fitness landscape, which is quantified in Fig. 2c, and the pairwise amino acid sequence identity are shown. **e** Pairwise amino acid sequence identity was computed using sequences found in the RBS base, namely at residues 135, 137, 145, 222, 225, and 226. **f** Pairwise amino acid sequence identity was computed using the sequences of the entire HA ectodomain. **e, f** Spearman's rank correlation ($\rho$) is indicated, with the $p$ value ($P$) stated in the parentheses. The raw data of the pairwise amino acid sequence identity are shown in Supplementary Fig. 8. Source data are provided as a Source Data file.

Those H-bonds between HA RBS and Gal-2 lead to a slight rotation of Gal-2 and all subsequent moieties toward the 190-helix (Fig. 5b). This rotation of Gal-2 is also made possible by mutation L226I, which lowers the floor of the RBS to create space for the Gal-2 O4 atom to rotate toward the base of the RBS (Fig. 5a, b). Of note, all of the mutations mentioned above do not belong to antigenic site B. Mutations G135T and N137S are located in antigenic site A, whereas W222R, G225N, and L226I are in antigenic site D[2].

Overall, this analysis demonstrates that natural mutations in the base of RBS can influence the conformation and mode of binding of the receptor. Since antigenic site B also interacts with the receptor (Fig. 1b), changes in receptor conformation will alter the evolutionary constraints of antigenic site B. This finding is consistent with our previous study, which showed that the amino acid preference at residue 190 is influenced by the receptor conformation[13]. In addition, the receptor specificity of human H3N2 viruses has been shown to be constantly evolving[34].

To further understand how the amino acid sequence in the base of the RBS may influence the local fitness landscape of antigenic site B, we compared the pairwise amino acid identity in the base of the RBS and the pairwise correlation of the local fitness landscape of antigenic site B among different genetic backgrounds. Here the base of the RBS is defined as residues 135, 137, 145, 222, 225, and 226. The amino acid sequence of the base of the RBS for each of the strains of interest is shown in Supplementary Fig. 8a, and their pairwise amino acid sequence identity is listed in Supplementary Fig. 8b. The similarity in the local fitness landscape of antigenic site B among different genetic backgrounds, which was quantified in Fig. 2c, correlated well with the pairwise amino acid sequence identity of the base of the RBS (Fig. 5e, Spearman's rank correlation = 0.66, p value = 0.007). In contrast, the similarity in the local fitness landscape of antigenic site B among different genetic backgrounds did not have significant correlation with the pairwise amino acid sequence identity of the entire HA ectodomain (Fig. 5e and Supplementary Fig. 8c, Spearman's rank correlation = 0.19, p value = 0.49). This analysis substantiates the notion that the amino acid sequence at the base of the RBS can modulate the local fitness landscape of antigenic site B.

## Discussion

The evolution of antigenic site B is functionally constrained owing to its significant overlap with the RBS. Subsequently, antigenic site B can only tolerate mutations that do not negatively impact receptor binding. This study demonstrates that evolutionary entrenchment and contingency are highly prevalent in antigenic site B. Such observations indicate that the evolutionary constraints of antigenic site B have been changing during the course of human H3N2 virus evolution. Concomitantly, the receptor-binding mode of human H3N2 viruses has been evolving (Fig. 5). With constantly changing evolutionary constraints, antigenic site B is able to continue exploring new sequence space that was not available in ancestral strains (evolutionary contingency) to promote antigenic drift. In other words, while the number of tolerable mutants in antigenic site B is limited for any given human H3N2 strain, which mutations are tolerable differ in different human H3N2 strains. Overall, the evolution of the local fitness landscape of antigenic site B expands the sequence space that is available for antigenic drift, despite the presence of strong functional constraints.

Predicting influenza evolution represents one of the keys to eradicate or control influenza virus. While methodologies for predicting evolution of influenza virus at a resolution of individual clades are being actively developed[35,36], it seems extremely difficult to predict the evolution of influenza virus at amino acid resolution. A previous study has shown that selection of antigenically advanced variants under antibody pressure can help predict the emerging mutations in the next antigenic drift[37]. However, our work here shows that, in antigenic site B, a mutation that has high fitness in one strain may be highly deleterious in another strain. The local fitness landscape of antigenic site B has been changing extensively throughout the evolution of human H3N2 viruses over the past 50 years and can be totally uncorrelated between strains that have been isolated far apart in time (e.g., Bris07 vs HK68, Fig. 2c). This observation implies that the local fitness landscape of antigenic site B is strongly influenced by the amino acid sequence in other parts of the HA protein. As a result, a mutation that drives antigenic drift in one strain may not even be tolerable in another strain due to epistasis. While short-term prediction of emerging mutations may be achievable, this study indicates that their prediction over a longer time frame of influenza evolution can be extremely difficult. In

the future, prediction of antigenic drift mutation might benefit from identifying epistatic interactions that involve both naturally observed and so far unobserved amino acid variants in all major antigenic sites.

## Methods

**Recombinant influenza virus**. All H3N2 viruses generated in this study were based on the influenza A/WSN/33 (H1N1) eight-plasmid reverse genetics system[38]. Chimeric 6:2 reassortments were employed with the HA ectodomains from the H3N2 strains of interest and the entire neuraminidase (NA) coding region from A/Hong Kong/1/1968 (HK68)[18]. For HA, the ectodomain was from HK68, whereas the non-coding region, N-terminal secretion signal, C-terminal transmembrane domain, and cytoplasmic tail were from A/WSN/33. For NA, the entire coding region was from HK68, whereas the non-coding region of NA was from A/WSN/33. The sequences of the HA ectodomains are indicated in Supplementary Data 1.

**Mutant library construction**. Mutant libraries were generated by a ligation strategy. For each mutant library, an insert and a vector were generated by PCR using KOD DNA polymerase (EMD Millipore) according to the manufacturer's instructions. The HA-encoding plasmid (pHW2000-HA) of the influenza eight-plasmid reverse genetics system was used as the template[38]. Primers for the insert contained the combinatorial mutations of interest and are shown in Supplementary Table 2. For insert PCR, forward primers a, b, c, and d (Supplementary Table 2) were mixed at equal molar ratio, whereas reverse primers a and b (Supplementary Table 2) were mixed at a molar ratio of 2:1. Primers for vector amplification are shown in Supplementary Table 2. Both the vector and insert were digested by BsmBI (New England Biolabs) and ligated using T4 DNA ligase (New England Biolabs). The ligated product was transformed into MegaX DH10B T1R cells (Life Technologies). At least one million colonies were collected. Plasmid mutant libraries were purified from the bacteria colonies using Maxiprep Plasmid Purification (Clontech Laboratories).

**Deep mutational scanning**. For each mutant library, transfection was performed in HEK 293T/MDCK-SIAT1 cells (Sigma-Aldrich, catalog number: 05071502-1VL) co-culture (ratio of 6:1) at 60% confluence in a T75 flask (75 cm$^2$) using lipofectamine 2000 (Life Technologies) according to the manufacturer's instructions. At 24 h post-transfection, cells were washed twice with phosphate-buffered saline (PBS), and cell culture medium was replaced with OPTI-MEM medium supplemented with 0.8 µg mL$^{-1}$ TPCK-trypsin. Virus mutant library was harvested at 72 h post-transfection and stored at −80 °C until used. For measuring virus titer by TCID$_{50}$ assay, MDCK-SIAT1 cells were washed twice with PBS prior to the addition of virus, and OPTI-MEM medium was supplemented with 0.8 µg mL$^{-1}$ TPCK-trypsin.

To passage the virus mutant libraries, MDCK-SIAT1 cells at 80% confluence in a T75 flask were washed twice with PBS and infected with a multiplicity of infection (MOI) of 0.01 in OPTI-MEM medium supplemented with 0.8 µg mL$^{-1}$ TPCK-trypsin. At 2 h post-infection, infected cells were washed twice with PBS and fresh OPTI-MEM medium supplemented with 0.8 µg mL$^{-1}$ TPCK-trypsin was added to the cells. At 24 h post-infection, supernatant containing the post-selection virus mutant library was harvested. Each replicate was transfected and passaged independently. Viral RNA was then extracted from the supernatant using the QIAamp Viral RNA Mini Kit (Qiagen Sciences, Germantown, MD). The extracted RNA was then reverse transcribed to cDNA using Superscript III reverse transcriptase (Life Technologies). The plasmid or the cDNA from the post-selection viral mutant libraries was amplified by PCR to add part of the adapter sequence required for Illumina sequencing using the primers listed in Supplementary Table 2.

A second PCR was performed to add the rest of the adapter sequence and index to the amplicon using primers: 5'-AAT GAT ACG GCG ACC ACC GAG ATC TAC ACT CTT TCC CTA CAC GAC GCT-3' and 5'-CAA GCA GAA GAC GGC ATA CGA GAT XXX XXX GTG ACT GGA GTT CAG ACG TGT GCT-3'. Positions annotated by an X represented the nucleotides for the index sequence. The final PCR products were submitted for next-generation sequencing on one lane of Illumina MiSeq PE300.

**Sequencing data analysis**. Sequencing data were obtained in FASTQ format and were parsed using SeqIO module in BioPython[39]. After trimming the primer sequences, both forward- and reverse-complemented reverse reads were translated into protein sequences. A paired-end read was then filtered and removed if the protein sequence translated from the forward read and that translated from the reverse-complemented reverse read did not match. Amino acids at the residues of interest were then extracted. The number of reads corresponding to each of the 576 variants was counted. For any given variant, input count represents the number of reads corresponding to the given variant in the plasmid mutant library, whereas selected count represents the number of reads corresponding to the given variant in the post-selection virus mutant library.

**Estimation of variant fitness, uncertainty, and preference**. The unnormalized fitness of each variant $k$ was estimated as $f_k = \log \frac{\text{sel}_k}{\text{in}_k}$, where $\text{sel}_k$ and $\text{in}_k$ are selected and input (before selection) counts. A pseudocount of 1 was added to the counts to estimate the (very low) fitness of variants with $\text{sel}_k = 0$. From the replicates of $\text{sel}_k$, we computed mean and variance of each variant. We found substantial deviation from the Poisson expectation that the mean and variance are equal (Supplementary Fig. 9). Therefore, we fit the data to a negative binomial distribution, which is commonly used to model overdispersed count data[40], with maximum likelihood, resulting in an overdispersion parameter $r = 23.8$, which leads to an estimate of fitness variance $\sigma_k^2 = \frac{1}{\text{sel}_k} + \frac{1}{\text{in}_k} + \frac{2}{r}$. Variant preferences are defined to be the fitness of each variant, normalized to have zero mean and unit standard deviation for each strain. The preference and unnormalized fitness of each variant are listed in Supplementary Data 2.

**Modeling fitness and decomposition of interactions**. We model fitness as a non-linear function of a sum of additive and pairwise effects. We chose a non-linear sigmoidal function[41], since they often describe biophysical processes and can account for non-specific interactions between sites[42]. Indeed, we find that a model without non-linearity has many more pairwise interactions compared to a model with non-linearity (Supplementary Fig. 10). This finding indicates that the pairwise interactions in the model with non-linearity are more likely to be a result of specific interactions. First, each site is encoded as a categorical variable with effects coding. This is also known as a zero-sum gauge and has the advantage that estimated effects are relative to the mean of the (included) amino acids and not one reference or WT sequence. The sum of additive and pairwise effects is

$$\phi(s(k)) = \sum_i h_i(s_i(k)) + \sum_{ij} J_{ij}(s_i(k), s_j(k))$$

where $s_i(k)$ is the amino acid at position $i$ of variant $k$. The zero-sum gauge means that the sum over $\sum_a h_i(a) = 0$ and $\sum_a J_{ij}(a, b) = 0$ for all positions. Fitness is then modeled as a non-linear function of the sum $g(\phi)$. We use a generalized logistic function,

$$g(\phi) = p_1 + e^{p_2}(1 + e^{\phi})^{p_3}$$

where $p_1$, $p_2$, and $p_3$ are free parameters that determine the shape. These parameters, along with the $h_i(a)$ and $J_{ij}(a, b)$, are estimated by minimizing the sum of the squared errors, weighted by the inverse of fitness variance,

$$\text{SSE} = \sum_k \left[ 1/\sigma_k^2 (f_k - g(\phi(s(k))))^2 \right]$$

where $s(k)$ is sequence of variant $k$. We estimate the confidence interval of the estimated additive and epistatic parameters by a parametric bootstrap: bootstrapped training data are generated by a normal distribution with mean equal to the fitness and variance equal to the fitness variance, then parameters are estimated for each bootstrap and the 95% confidence interval is calculated from an ensemble of 100 bootstraps.

**Visualization of the local fitness landscapes**. Sequence logos in Fig. 2a were generated by WebLogo[43]. Variant networks in Supplementary Fig. 3 were generated by Graphviz.

**Analysis of natural sequences**. A total of 45,218 full-length HA protein sequences from human H3N2 were downloaded from the Global Initiative for Sharing Avian Influenza Data (http://gisaid.org)[44] (Supplementary Table 3). Amino acid sequences of HA residues 156, 158, 159, 190, 193, and 196 in individual strains were extracted. Individual sequences were grouped by year of isolation and their normalized preferences in different genetic backgrounds were plotted in Fig. 3a. The human H3N2 HA protein sequences used in this study are listed in Supplementary Data 3.

**Calculation of KL divergence**. We evaluate the sequence divergence between the ensembles of natural sequences collected in different years, using the residue averaged symmetric KL divergence in each antigenic sequence region. We denote the frequency of amino acid $a$ at position $i$ among sequences collected at a given year $t$ by $x_i^\alpha(t)$. The residue averaged symmetric KL divergence $D_{\text{KL}}$ for a specific region of length $L$ follows,

$$D_{\text{KL}}(t, t') = \frac{1}{2L} \sum_{\alpha=1}^{20} \sum_{i=1}^{L} x_i^\alpha(t) \log \frac{x_i^\alpha(t)}{y_i^\alpha(t, t')} + x_i^\alpha(t') \log \frac{x_i^\alpha(t')}{y_i^\alpha(t, t')}$$

where $y_i^\alpha(t, t') = [x_i^\alpha(t) + x_i^\alpha(t')]/2$. The symmetric KL divergence is a non-negative measure of distance between distributions and a larger $D_{\text{KL}}(t, t')$ indicates a larger divergence between the sequences collected in years $t$ and $t'$. To estimate the uncertainty due to finite sampling, we provide a 95% confidence interval for the KL divergence based on 100 parametric bootstraps of a multinomial distribution for the estimated frequencies at each position.

**Crystallization and structural determination**. The HA68 HA ectodomain, which contains HA1 residues 11–329 and HA2 residues 1–176, was fused to an N-terminal gp67 signal peptide and to a C-terminal biotinylation site, trimerization domain, and 6x His-tag. Recombinant bacmid DNA that carried the HK68 HA ectodomain construct was generated using the Bac-to-Bac system (Life Technologies). Baculovirus was generated by transfecting purified bacmid DNA into Sf9 cells (ATCC, catalog number: CRL-1711) using FuGene HD (Promega). HK68 HA ectodomain was expressed by infecting suspension cultures of High Five cells (Thermo Fisher Scientific, catalog number: B85502) with baculovirus at an MOI of 5–10 and incubating at 28 °C shaking at 110 r.p.m. for 72 h. The supernatant was concentrated. HA0 was purified by Ni-NTA, buffer exchanged into 10 mM Tris-HCl [pH 8.0] and 50 mM NaCl, and concentrated to 10 mg mL$^{-1}$. Initial crystal screening was carried out using our high-throughput, robotic CrystalMation system (Rigaku) at TSRI. Diffraction-quality crystals were obtained from 0.1 M sodium cacodylate [pH 6.5], 6% PEG 8000, and 39% 2-methyl-2,4-pentanediol at 20 °C by sitting drop vapor diffusion method with 500 µL reservoir solution and each drop consisting 0.8 µL protein + 0.8 µL precipitant. The resulting crystals were soaked in 20 mM 6′SLNLN for 2 h, flash cooled, and stored in liquid nitrogen until data collection.

Diffraction data were collected at the APS GM/CA-CAT 23ID-B. The data were indexed and integrated and scaled using HKL2000 (HKL Research)[45]. The structure was solved by molecular replacement using Phaser[46] with PDB: 6AOR[32] as the molecular replacement model, and the structure was rebuilt using Coot[47] and refined using Refmac5[48]. Ramachandran statistics were calculated using MolProbity[49].

**Reporting summary**. Further information on research design is available in the Nature Research Reporting Summary linked to this article.

## Data availability

Raw sequencing data have been submitted to the NIH Short Read Archive under accession number: BioProject PRJNA563320. The X-ray coordinates and structure factors have been deposited to the RCSB Protein Data Bank under accession code: 6TZB. The source data underlying Figs. 3a, b, 4a–c, and 5e, f are provided as a Source Data file. All of the other data that support the conclusions of the study are available from the corresponding author upon request.

## Code availability

Custom python scripts for analyzing the deep mutational scanning data have been deposited to https://github.com/wchnicholas/site_B_landscape.

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

## Acknowledgements

We thank Steven Head and Jessica Ledesma at TSRI Next Generation Sequencing Core and Marta Łuksza and Jesse Bloom for helpful discussions. This work was supported by National Institutes of Health (NIH) R56 AI127371, R01 AI114730, K99 AI139445, Bill and Melinda Gates Foundation OPP1170236, DFG grant (SFB1310) for Predictability in Evolution, and the MPRG funding through the Max Planck Society.

## Author contributions

N.C.W., J.O., A.J.T., A.N., and I.A.W. conceived and designed the study; N.C.W. performed the deep mutational scanning experiment, collected the X-ray data and determined and refined the X-ray structures. N.C.W., J.O., A.J.T., and A.N. analyzed the data. C.M.N. synthesized the 6′SLNLN; N.C.W. and I.A.W. wrote the paper and all authors reviewed and edited the paper.

## Competing interests

The authors declare no competing interests.
