## [Peer Review File · Nature Communications]

Reviewers' Comments:

Reviewer #1:

Remarks to the Author:

In this paper, the authors study the evolution over time of the local fitness landscape of hemagglutinin antigenic site B (AgsB) of influenza H3N2 using deep mutational scanning. Focusing on AgsB makes sense as it involves receptor binding sites and has recently become immunodominant. Specifically, they focused on six residues within AgsB, which have reached high frequencies during H3N2 evolution since 1968. From this study, the authors found that the local fitness landscape of AgsB highly depends on the genetic background, i.e., other residues in HA, and shows signature of evolutionary entrenchment and contingency. This is suggestive of an evolving fitness landscape of influenza (also referred to as "seascape" by other researchers) under the strong influence of natural and vaccine-induced immune pressure. They also performed structural analysis to demonstrate that the receptor binding mode of H3N2 has also evolved over time, providing further escape pathways to the virus.

While I think the theory that the fitness landscape of influenza is evolving over time has been known (works by Michael Lassig and Richard Neher), a systematic analysis proving it to be so, to the best of my knowledge, has never been done before. Specifically, I appreciate the research approach of the authors, i.e., considering a very simple system (six residues of HA AgsB) and performing an in-depth analysis on the fitness of all their possible variants in six different backgrounds (from 1968 to 2016) to provide insights related to an interesting biological problem.

Overall, I think this is a very good work and the results are quite interesting. However, some parts of the paper need further elaboration. The authors should address the following comments for improving the manuscript so that the work is comprehensible for the general audience of this journal.

Main comments:

1. I understand that the deep mutational scanning (DMS) method has been widely used to study the fitness landscape of viruses. However, I think it is important to summarize the DMS method in simple language in the Results section so that the wide readership of Nature Communications can comprehend it. For example, the authors should explain why the previous DMS and influenza HA related reports did not study epistatic interactions, were there any methodological limitation, how have they incorporated it in their current method, how did they make sure that mutations appear only in the six targeted residues and not in the background strain, did they use any type of filtering after passaging, etc.

2. The Methods part needs to be revised and more details should be provided. For example,

- i) Lines 349-351: There seems to be a disconnect at the start of this statistical analysis part from the experimental part described in the previous section.
- ii) Line 351: It is not clear what the selected and input counts mean. Elaborating or explaining with an example would help here.
- iii) Lines 352-353: Provide references for the specific measurement noise model used. How robust are the results to the choice of this noise model?
- iv) Lines 366-367: Provide references for the specific non-linear function used to model fitness. And again, how robust are the obtained results to the choice of this specific non-linear model?

3. In Fig. 3, the calculation of KL divergence involves computation of the frequency of a particular amino acid at a protein position among sequences collected in a particular year. As the sampling of influenza sequences over the years has been highly non-uniform (e.g., there are very few sequences

from 1968 as compared to those in recent years), did the authors account for this sampling bias while calculating KL divergence? If not, how do results change by accounting for it.

4. Lines 186-187: Regarding the statement "A fitness landscape can be approximated by additive fitness and pairwise epistatic effects, which are easier to interpret than the full landscape", I don't think this is true and depends a lot on the underlying structure of the fitness landscape being studied. The authors need to elaborate this point. Also, please provide references to support this statement.

5. Line 199: I agree with the general comment that epistasis promotes diversity of functional sequence space. However, given their specific non-linear fitness model, can the authors elaborate how positive and negative epistatic terms can promote diversity?

6. Lines 413-414: I hope that the authors will also share amino acid sequences and the associated preferences/fitness values in all six backgrounds with the scientific community. This will enable the interested researchers to make use of the paper findings easily.

Minor comments:

1. Line 75: Instead of giving a number 576 which the reader would not understand at this point, it would be better to replace it with something like "all possible variants".
2. Lines 81-82: Describe what evolutionary entrenchment and contingency mean here (when these are first mentioned), instead of describing them later in the Results section.
3. Lines 91-96: Is it possible to study the variants comprising non-observed amino acids at the six studied residues in AgsB? Such a study might be useful for understanding potential future virus escape pathways (fit variants) within AgsB in a recent strain. I think it would be very interesting from a vaccine design perspective. The authors may discuss about this and amplify their concluding statements in the Discussion section.
4. Line 206: Mention that there are 133 parameters after one ignores the pairs between different amino acids at the same residue.
5. Indicate residue numbers on the sequence logos in Fig. 2.

Reviewer #2:

Remarks to the Author:

SUMMARY

In this study, the authors measure the replication fitness a combinatorial library of variants that have reached high frequency in natural sequences at 6 sites in antigenic site B of H3 hemagglutinin (576 variants in total), in 6 genetic backgrounds to understand how epistasis affects the evolution of antigenic site B. The authors find the correlations of preferences for each variant is high between some genetic backgrounds, while very low between others. The authors also examine the preferences of naturally isolated haplotypes comprising the 6 studied sites in each of the genetic backgrounds, and find that as time since sequence isolation and sequence diversity increases, the preference of that haplotype tends to decrease in a given genetic background. They also examined the contributions of additive and pairwise fitness effects to each of the local fitness landscapes. Finally, the authors solved the structure of HK68 HA in complex with a receptor analog, 6'SLNLN, and compared this structure to that of the previously-solved Bris07-6'-SLNLN to understand some of the mechanisms underlying the observed epistatic interactions in antigenic site B.

This was a very interesting study with important implications for the field, and we strongly support its publication. However, there are several points that we think the authors should address in revisions

prior to publication:

MAJOR COMMENTS

* The concept behind Figure 3 is very interesting, however, it is poorly explained in the text. Figure S4 does a poor job of explaining the idea behind the analysis, or how it was done. Importantly, how many protein sequences per year were chosen, and how were they sampled?

- In addition, how frequently are each of the possible 576 genotypes observed in nature? How does the distribution of these haplotypes change over time?

* We suggest the authors consider moving Figures S2 and S3 to the main text. They can be incorporated as panels into Figure 2. The network diagrams from Figure 2 could be moved to the supplement, as they are difficult to interpret and do not contribute significant additional information to the interpretation of the correlations between genetic backgrounds or the logo plots.

* I'm not sure if "normalized preference" is the best metric to use for these data. The normalized preference sets the mean preference to zero for each genetic background. However, a normalized preference of zero does not have any biological meaning as to whether a genotype would be fit in nature. Perhaps it would be better to calculate fitness effects relative to the "wild type" sequence for each background, as it is at least known that each wild type sequence surpasses a fitness threshold.

- Related, Figure S5: The solid black line at 0 suggests that a preference of "0" has a biological meaning, such as a fitness threshold for viability, but it does not. 0 is simply the average preference of all 576 genotypes in a given genetic background.

* In lines 211-216, the authors state that the epistatic effects are more conserved than the additive effects. This is true for their dataset, but it's very important to note that their analysis is conditioned on amino acids that actually fixed at some point during HA evolution. Presumably if the authors looked at all amino acids at these sites, the additive effects might dominate as many amino acids could be highly unfavorable at the sites. In other words, the relative importance of additive and epistatic effects probably depends on what mutations are being tested—and here the authors are only testing mutations that are well tolerated in at least one actual background. This distinction needs to be made clear in the discussion of the relevant results.

* Accessibility of data and analysis code:

- A statement that the scripts will be available on Github is not sufficient; please include the actual URL where the repository can be accessed. Likewise for the deep sequencing data.

- There is insufficient detail in the methods and results about how the naturally-occurring sequences were sampled.

- A file with the full HA sequences of the sampled naturally occurring samples (Fig. 3) should be included in the supplement, as well as a clear explanation regarding how these sequences were sampled.

MINOR COMMENTS

* Figure 2: Please number each site in each logo plot, and include labels with the wild type residue at each site in each background.

* The mathematical formulas in lines 363-370 are complex enough that they should be formatted as equations.

* In Figure 2, S1, S2, and S3, please highlight the points that correspond to the wild type variant in each background. This could be done as each of the 6 genotypes highlighted in a different color, so the reader can see how preferred the wild type genotype is in its natural genetic background, as well

in each of the other 5 genetic backgrounds.

* Figure 4: the text describing “additive fitness effect” vs. “pairwise epistatic effect” is very unclear. If we did not already understand these concepts, we wouldn't have been able to get it from the current text. These concepts should be explained in much greater detail, with references to relevant literature, in the main results section of the text.

* First sentence of abstract: Is it really correct to say that evolvability “fuels” the evolution? Evolvability is a pre-requisite for the evolution, but selection really fuels it.

* Line 45: “Subsequently” doesn't seem to be the correct word here. Maybe “Consequently” is better?

* Line 102: should be “was” the vaccine strain, not “is” the vaccine strain.

* The methods should better clarify the origin of the non-ectodomain regions of HA and NA. Were these from WSN?

* Why are Bk79 and Bei89 so highly correlated? Are these two strains more genetically or structurally similar, particularly in the antigenic region B location in question.

AUTHOR IDENTITIES:

We have opted into the options to publish the review and reveal our identities. In accordance with the online reviewer instructions, we are also adding our names here to the end of the review: Allie Greaney and Jesse Bloom.

Reviewer #3:

Remarks to the Author:

In this manuscript, “Major hemagglutinin antigenic site B of human influenza H3N2 viruses has an evolving local fitness landscape.”, Wu and colleagues build off of their previous work analyzing the fitness and mutational landscape of the influenza hemagglutinin, in particular the conserved receptor binding site (RBS) (Wu et al Nat Coms 2018). Specifically, they selected six residues in antigenic site B and performed deep mutational scanning of historical H3N2 viruses. These historical were chosen because they are ~10yrs apart and were also vaccine components. (Unintentionally perhaps, 4 (HK68, BK79, BE89 and BR07) of the 6 strains are also representative strains that define antigenic clusters from Bedford, Smith and colleagues (eLife,2014; 3:e0194)). The replication fitness of 576 variants were tested, and perhaps not surprisingly, the “local” fitness landscape depends on the genetic background (i.e., the specific HA strain). Comparing their previously determined BR07

Minor:

1) Perhaps modify Figure 1 to show the other major antigenic sites on the HA head near site B and RBS. How much overlap (calculated surface area and number of amino acids) is there between these sites?

2) The authors state that six residues were selected at Antigenic site B because they “reached high occurrence frequency during the natural evolution of H3N2 viruses”; how was this determined / what do the authors mean by this? How do these six residues compare to other residues within antigenic site B in terms of their “occurrence frequency”? Could the authors explain in a little more detail how they arrived at these specific six residues?

3) Were any of the the variants that have reduced fitness tested for affinity for sialic acid? Do the authors feel this would further support their arguments on functional constraint of antigenic site B and the RBS?

4) Supp Fig 7: consider modifying to stereo view for ease of reader interpreting electron density.

Response to reviewers

Reviewer #1 (Remarks to the Author):

In this paper, the authors study the evolution over time of the local fitness landscape of hemagglutinin antigenic site B (AgsB) of influenza H3N2 using deep mutational scanning. Focusing on AgsB makes sense as it involves receptor binding sites and has recently become immunodominant. Specifically, they focused on six residues within AgsB, which have reached high frequencies during H3N2 evolution since 1968. From this study, the authors found that the local fitness landscape of AgsB highly depends on the genetic background, i.e., other residues in HA, and shows signature of evolutionary entrenchment and contingency. This is suggestive of an evolving fitness landscape of influenza (also referred to as “seascape” by other researchers) under the strong influence of natural and vaccine-induced immune pressure. They also performed structural analysis to demonstrate that the receptor binding mode of H3N2 has also evolved over time, providing further escape pathways to the virus.

While I think the theory that the fitness landscape of influenza is evolving over time has been known (works by Michael Lassig and Richard Neher), a systematic analysis proving it to be so, to the best of my knowledge, has never been done before. Specifically, I appreciate the research approach of the authors, i.e., considering a very simple system (six residues of HA AgsB) and performing an in-depth analysis on the fitness of all their possible variants in six different backgrounds (from 1968 to 2016) to provide insights related to an interesting biological problem.

Overall, I think this is a very good work and the results are quite interesting. However, some parts of the paper need further elaboration. The authors should address the following comments for improving the manuscript so that the work is comprehensible for the general audience of this journal.

Response: Thank you for the positive and constructive comments, which have helped us improve the manuscript.

Main comments:

1. I understand that the deep mutational scanning (DMS) method has been widely used to study the fitness landscape of viruses. However, I think it is important to summarize the DMS method in simple language in the Results section so that the wide readership of Nature Communications can comprehend it. For example, the authors should explain why the previous DMS and influenza HA related reports did not study epistatic interactions, were there any methodological limitation, how have they incorporated it in their current method, how did they make sure that mutations appear only in the six targeted residues and not in the background strain, did they use any type of filtering after passaging, etc.

Response: Thank you for the suggestion. We have now included a summary of the deep mutational scanning method in results section of the revised manuscript:

“Deep mutational scanning involves construction of a mutant library and measurement of the frequency change of individual mutants in the mutant library during a specified selection using next-generation sequencing. For each mutant,

the magnitude of the frequency change is the proxy of its phenotype. Most deep mutational scanning studies, including those on influenza HA^{14, 16, 17}, have analyzed mutant libraries that mainly consisted of a single mutant. As a result, epistatic interactions could not be studied since mutants with two or more mutations were rare in these mutant libraries. Here, using PCR primers that carry degenerate nucleotides, we were able to precisely introduce the desired set of amino-acid mutations into the six residues of interest (see Methods).”

2. The Methods part needs to be revised and more details should be provided. For example,

i) Lines 349-351: There seems to be a disconnect at the start of this statistical analysis part from the experimental part described in the previous section.

Response: In methods section of the revised manuscript, a subsection “Sequencing data analysis” is added, which improves the connection between the statistical analysis part and the experimental part:

“Sequencing data were obtained in FASTQ format and were parsed using SeqIO module in BioPython³⁹. After trimming the primer sequences, both forward and reverse-complemented reverse reads were translated into protein sequences. A paired-end read was then filtered and removed if the protein sequence translated from the forward read and that translated from the reverse-complemented reverse read did not match. Amino acids at the residues of interest were then extracted. The number of reads corresponding to each of the 576 variants was counted. For any given variant, input count represents the number of reads corresponding to the given variant in the plasmid mutant library, whereas selected count represents the number of reads corresponding to the given variant in the post-selection virus mutant library.”

ii) Line 351: It is not clear what the selected and input counts mean. Elaborating or explaining with an example would help here.

Response: Thanks – this is now clarified in the newly added subsection “Sequencing data analysis” of the methods section in the revised manuscript:

“For any given variant, input count represents the number of reads corresponding to the given variant in the plasmid mutant library, whereas selected count represents the number of reads corresponding to the given variant in the post-selection virus mutant library.”

iii) Lines 352-353: Provide references for the specific measurement noise model used. How robust are the results to the choice of this noise model?

Response: In the revised manuscript, we have added a reference for the negative binomial distribution model:

“Therefore, we fit the data to a negative binomial distribution, commonly used to model overdispersed count data⁴⁰”

In general, the maximum likelihood parameters of a model depend on the form of the likelihood function (the noise model). We should not expect robustness under different likelihoods unless they are effectively the same. We chose a model based on our knowledge of the experimental noise. A Poisson approximation is usually appropriate for count data. However, we saw substantial overdispersion in the count data, meaning the mean and variance of replicate counts are not equal; this is now shown in Supplementary Fig. 9. This finding led us to modify the estimate of the fitness variance by including a term, which accounts for this overdispersion based on fitting a negative binomial. We now discuss all of these points more clearly in the methods subsection, “Estimation of variant fitness, uncertainty, and preference”.

To address the question about sensitivity of the results, we also fitted another model, which uses fitness variances from a Poisson approximation, and found the additive and pairwise parameters to be nearly identical (correlation = 0.999). However, the negative binomial is a better model of the experimental noise, and leads to better estimates of the uncertainty in the parameters, so that is why we present it in the manuscript.

iv) Lines 366-367: Provide references for the specific non-linear function used to model fitness. And again, how robust are the obtained results to the choice of this specific non-linear model?

Response: We have added a reference for the non-linear function:

Nelder JA. The fitting of a generalization of the logistic curve. *Biometrics* 17, 89-110 (1961).

Again, we should not expect robustness of the inferred additive and pairwise interactions with the choice of non-linear function, as this changes the form of the likelihood.

We chose to add a non-linear function on top of the additive and pairwise effects, to account for epistasis between sites that are due to non-specific interactions, as described in:

Otwinowski J, McCandlish DM, Plotkin JB. Inferring the shape of global epistasis. *Proc Natl Acad Sci U S A* 115, E7550-E7558 (2018).

Indeed, we found that a model without the global non-linearity has more significant pairwise interactions. This indicates that many of these pairwise interactions are likely to be non-specific interactions that arise from a global non-linearity. We have added a figure to the SI with a caption that discusses this point and modified the text in the methods section:

“We chose a non-linear sigmoidal function⁴¹, since they often describe biophysical processes, and can account for non-specific interactions between sites⁴². Indeed, we find that a model without non-linearity has many more pairwise interactions compared to a model with non-linearity (Supplementary Fig. 10). This finding indicates that the pairwise interactions in the model with non-linearity are more likely to be a result of specific interactions.”

The choice of specific non-linear function is also important and, in general, can impact the resulting additive and pairwise effects. Different classes of non-linear functions

reflect different assumptions put into a model. Ultimately, we have to make a choice, and we chose the generalized logistic curve that is flexible and has relatively few parameters. Moreover, sigmoidal functions often describe biophysical processes, which are likely to introduce such non-specific interactions between sites.

3. In Fig. 3, the calculation of KL divergence involves computation of the frequency of a particular amino acid at a protein position among sequences collected in a particular year. As the sampling of influenza sequences over the years has been highly non-uniform (e.g., there are very few sequences from 1968 as compared to those in recent years), did the authors account for this sampling bias while calculating KL divergence? If not, how do results change by accounting for it.

Response: Thank you for pointing this out. We now provide 95% confidence intervals for the estimated KL divergence based on a parametric bootstrap of the underlying multinomial distribution for the amino acid frequencies at each position. This then quantifies the uncertainty due to fewer samples in earlier years. We also now use a symmetric KL divergence, which is a more robust estimate for multinomial distributions. Nevertheless, these new results in the revised manuscript are almost identical to what we showed previously. We have also added error bars in Fig. 3 to show the 95% confidence interval.

4. Lines 186-187: Regarding the statement “A fitness landscape can be approximated by additive fitness and pairwise epistatic effects, which are easier to interpret than the full landscape”, I don’t think this is true and depends a lot on the underlying structure of the fitness landscape being studied. The authors need to elaborate this point. Also, please provide references to support this statement.

Response: In the revised manuscript, we have elaborated on why we choose to approximate the fitness landscape by additive fitness effect and pairwise epistatic effect:

“Fitness landscapes of molecules are often complex with both specific and non-specific interactions²⁶. However, interpreting the structure of these high-dimensional landscapes is difficult²⁷ and highly sensitive to experimental noise. Previous work has shown that approximate fitness landscapes with additive and pairwise interactions among residues can accurately describe biophysical properties of proteins, including protein residue-residue contacts^{28, 29}. Additive fitness effects describe the independent contributions of each amino-acid variant to fitness, whereas pairwise epistatic interaction effect describes how pairs of amino acids synergistically (positively) or antagonistically (negatively) impact the fitness^{30, 31}. Here, we construct a statistical model to infer the additive fitness effects and pairwise epistatic interactions that underlie the six local fitness landscapes (see Methods). This model provides an interpretable description of the high-dimensional experimental landscape in the presence of the fitness measurement noise.”

5. Line 199: I agree with the general comment that epistasis promotes diversity of functional sequence space. However, given their specific non-linear fitness model, can the authors elaborate how positive and negative epistatic terms can promote diversity?

Response: As the reviewer has astutely pointed out, epistasis that restores function can lead to the survival of otherwise deleterious mutations in a population and, therefore,

could increase the diversity of the functional sequence space. The logistic form for a global nonlinearity in the fitness landscape that we infer can further hamper the effect of deleterious mutations.

In principle, the flattening of the logistic function at low fitness implies a reduction in the magnitude of the deleterious effects compared to the linear case. This slight boost in fitness could lead to further diversification of the functional sequence space. However, this flattening is likely to reflect the lower limits of detectable fitness in the experiments and, therefore, we are not confident enough to make a connection to diversity of the functional sequence space. We believe making any such connection would require further experimental data (collected at low fitness values) and also a detailed population genetics investigation of this phenomenon.

6. Lines 413-414: I hope that the authors will also share amino acid sequences and the associated preferences/fitness values in all six backgrounds with the scientific community. This will enable the interested researchers to make use of the paper findings easily.

Response: The amino acid sequence as well as the fitness and preference for each variant in each of the six backgrounds are listed in Supplementary Data 1. This is stated in the methods section of the revised manuscript:

“The preference and unnormalized fitness of each variant are listed in Supplementary Data 1.”

Minor comments:

1. Line 75: Instead of giving a number 576 which the reader would not understand at this point, it would be better to replace it with something like “all possible variants”.

Response: In the revised manuscript, we have changed that to “all possible variants” as suggested.

2. Lines 81-82: Describe what evolutionary entrenchment and contingency mean here (when these are first mentioned), instead of describing them later in the Results section.

Response: In the revised manuscript, the definitions of evolutionary entrenchment and contingency are described when they are first mentioned in the introduction.

3. Lines 91-96: Is it possible to study the variants comprising non-observed amino acids at the six studied residues in AgsB? Such a study might be useful for understanding potential future virus escape pathways (fit variants) within AgsB in a recent strain. I think it would be very interesting from a vaccine design perspective. The authors may discuss about this and amplify their concluding statements in the Discussion section.

Response: In the revised manuscript, we have added a sentence in the discussion section to discuss such possibility:

“In the future, prediction of antigenic drift mutation might benefit from identifying epistatic interactions that involves both naturally observed and non-observed amino-acid variants in all major antigenic sites.”

4. Line 206: Mention that there are 133 parameters after one ignores the pairs between different amino acids at the same residue.

Response: Thanks – we have added a sentence in the revised manuscript to describe this.

5. Indicate residue numbers on the sequence logos in Fig. 2.

Response: The residue numbers on the sequence logos in Fig. 2 are indicated in the revised manuscript.

Reviewer #2 (Remarks to the Author):

SUMMARY

In this study, the authors measure the replication fitness a combinatorial library of variants that have reached high frequency in natural sequences at 6 sites in antigenic site B of H3 hemagglutinin (576 variants in total), in 6 genetic backgrounds to understand how epistasis affects the evolution of antigenic site B. The authors find the correlations of preferences for each variant is high between some genetic backgrounds, while very low between others. The authors also examine the preferences of naturally isolated haplotypes comprising the 6 studied sites in each of the genetic backgrounds, and find that as time since sequence isolation and sequence diversity increases, the preference of that haplotype tends to decrease in a given genetic background. They also examined the contributions of additive and pairwise fitness effects to each of the local fitness landscapes. Finally, the authors solved the structure of HK68 HA in complex with a receptor analog, 6'SLNLN, and compared this structure to that of the previously-solved Bris07-6'-SLNLN to understand some of the mechanisms underlying the observed epistatic interactions in antigenic site B.

This was a very interesting study with important implications for the field, and we strongly support its publication. However, there are several points that we think the authors should address in revisions prior to publication:

Response: Thank you for the positive and constructive comments.

MAJOR COMMENTS

* The concept behind Figure 3 is very interesting, however, it is poorly explained in the text. Figure S4 does a poor job of explaining the idea behind the analysis, or how it was done. Importantly, how many protein sequences per year were chosen, and how were they sampled?

Response: We have revised Figure S4 as well as added two sentences in the results section to improve the explanation of the idea behind the analysis:

“Subsequently, we plotted the preference of individual natural sequence variants (haplotypes of residues 156, 158, 159, 190, 193, and 196) in different focal genetic backgrounds (HK68, Bk79, Bei89, Mos99, Bris07, and NDako16) against the year of strain isolation. This analysis tracked how natural sequence variants from different year would fit in a given focal genetic background.”

In the revised manuscript, the number of HA protein sequences that were chosen per year is listed in Table S3. Also, the analysis is described in the Methods section:

“A total of 45,218 full-length HA protein sequences from human H3N2 were downloaded from the Global Initiative for Sharing Avian Influenza Data (GISAID; <http://gisaid.org>)⁴⁴ (Supplementary Table 3). Amino-acid sequences of HA residues 156, 158, 159, 190, 193, and 196 in individual strains were extracted. Individual sequences were grouped by year of isolation and their normalized preferences in different genetic backgrounds were plotted in Fig. 3a. The human H3N2 HA protein sequences used in this study are listed in Supplementary Data 2.”

- In addition, how frequently are each of the possible 576 genotypes observed in nature? How does the distribution of these haplotypes change over time?

Response: Out of 576 possible haplotypes, 38 have been observed in nature. As suggested by the reviewer, the frequency of each of these 38 haplotypes over time has been plotted. This result is presented in Figure S1 and is mentioned in the results section:

“Out of these 576 variants, 38 of them have been observed in naturally circulating human H3N2 strains (Supplementary Fig. 1).”

* We suggest the authors consider moving Figures S2 and S3 to the main text. They can be incorporated as panels into Figure 2. The network diagrams from Figure 2 could be moved to the supplement, as they are difficult to interpret and do not contribute significant additional information to the interpretation of the correlations between genetic backgrounds or the logo plots.

Response: Thanks – we agree and have moved the network diagram to the supplement as Figure S3, and moved previous Figures S2 and S3 to Figures 2b and 2c, respectively, in the revised manuscript.

* I'm not sure if “normalized preference” is the best metric to use for these data. The normalized preference sets the mean preference to zero for each genetic background. However, a normalized preference of zero does not have any biological meaning as to whether a genotype would be fit in nature. Perhaps it would be better to calculate fitness effects relative to the “wild type” sequence for each background, as it is at least known that each wild type sequence surpasses a fitness threshold.

Response: The main reason why we did not calculate fitness effects relative to the wild type (WT) for each background was because the WT sequence for Mos99 was not included in our deep mutational scanning experiment. This point is now stated in the legend of Figure 2:

“Of note, the WT sequence of Mos99 contains a naturally rare variant T196. Therefore, the WT sequence of Mos99 was not included in our deep mutational scanning experiment.”

Moreover, our conclusion would remain the same with any linear-transformation of the result. In other words, computing the fitness effect either relative to the WT or by setting the mean preference to zero would not alter the conclusion of our study.

- Related, Figure S5: The solid black line at 0 suggests that a preference of “0” has a biological meaning, such as a fitness threshold for viability, but it does not. 0 is simply the average preference of all 576 genotypes in a given genetic background.

Response: We agree and have removed the solid black line at 0 in Figure S5 of the revised manuscript.

* In lines 211-216, the authors state that the epistatic effects are more conserved than the additive effects. This is true for their dataset, but it’s very important to note that their analysis is conditioned on amino acids that actually fixed at some point during HA evolution. Presumably if the authors looked at all amino acids at these sites, the additive effects might dominate as many amino acids could be highly unfavorable at the sites. In other words, the relative importance of additive and epistatic effects probably depends on what mutations are being tested—and here the authors are only testing mutations that are well tolerated in at least one actual background. This distinction needs to be made clear in the discussion of the relevant results.

Response: Thank you for the excellent comment. We agree that this should be pointed out. In the results section of the revised manuscript, we have specifically stated that our analysis has focused on major natural variants in antigenic site B:

“This analysis demonstrates that among major naturally observed amino-acid variants in antigenic site B, both the additive fitness effects and the pairwise epistatic effects have large fluctuation during the course of human H3N2 evolution to date.”

Of note, we noticed that there was a mistake in our previous code for this analysis – our apologies. We have fixed the mistake and adjusted the conclusion for this section accordingly. Specifically, we previously claimed that the epistatic effect is more conserved than the additive fitness effect, whereas the revised analysis has now demonstrated that both additive effect and epistatic effect are not conserved.

In the revised manuscript, we have also acknowledged that both the additive fitness effect and the pairwise epistatic effect may be more conserved if unobserved amino-acid variants are also included in our analysis:

“Nevertheless, we acknowledge that both the additive fitness effect and the pairwise epistatic effect are likely to be more conserved when unobserved amino-acid variants in antigenic site B are also considered in our analysis, since they might be deleterious regardless of the genetic backgrounds.”

* Accessibility of data and analysis code:

- A statement that the scripts will be available on Github is not sufficient; please include the actual URL where the repository can be accessed. Likewise for the deep sequencing data.

Response: The URL for the Github repository is provided in the revised manuscript:

“Custom python scripts for analyzing the deep mutational scanning data have been deposited to https://github.com/wchnicholas/site_B_landscape.”

The accession number of the deep sequencing data is also provided in the revised manuscript:

“Raw sequencing data have been submitted to the NIH Short Read Archive under accession number: BioProject PRJNA563320.”

- There is insufficient detail in the methods and results about how the naturally-occurring sequences were sampled.

Response: In the methods section of the revised manuscript, a subsection “Analysis of natural sequences” is added to detail the natural sequences:

“A total of 45,218 full-length HA protein sequences from human H3N2 were downloaded from the Global Initiative for Sharing Avian Influenza Data (GISAID; <http://gisaid.org>)³⁵ (Supplementary Table 3).”

The number of HA protein sequences per year are shown in Supplementary Table 3.

- A file with the full HA sequences of the sampled naturally occurring samples (Fig. 3) should be included in the supplement, as well as a clear explanation regarding how these sequences were sampled.

Response: In the revised manuscript, a clear description of the naturally occurring sequences is provided (see response above). We have also provided a file with full HA sequences of the naturally occurring sequences used in this study as Supplementary Data 2. This is stated in the subsection “Analysis of natural sequences” of the methods section in the revised manuscript:

“The human H3N2 HA protein sequences used in this study are listed in Supplementary Data 2.”

MINOR COMMENTS

* Figure 2: Please number each site in each logo plot, and include labels with the wild type residue at each site in each background.

Response: The residue numbers on the sequence logos in Fig. 2 are labeled in the revised manuscript. The wild type sequences in different backgrounds at the sites of interest are also indicated.

* The mathematical formulas in lines 363-370 are complex enough that they should be formatted as equations.

Response: Those mathematical formulas are formatted as equations in the revised manuscript.

* In Figure 2, S1, S2, and S3, please highlight the points that correspond to the wild type variant in each background. This could be done as each of the 6 genotypes highlighted in a different color, so the reader can see how preferred the wild type genotype is in its natural genetic background, as well in each of the other 5 genetic backgrounds.

Response: As suggested by the reviewer, the data points that correspond to the wild type variant in each background are highlighted in Figure 2b (previous Figure S2), Figure 2c (previous Figure S3), and Figure S2 (previous Figure S1).

* Figure 4: the text describing “additive fitness effect” vs. “pairwise epistatic effect” is very unclear. If we did not already understand these concepts, we wouldn’t have been able to get it from the current text. These concepts should be explained in much greater detail, with references to relevant literature, in the main results section of the text.

Response: In the revised manuscript, we have significantly expanded the part that describes additive fitness effect and pairwise epistatic effect, and how they are related to fitness landscape:

“Fitness landscapes of molecules are often complex with both specific and non-specific interactions²⁶. However, interpreting the structure of these high-dimensional landscapes is difficult²⁷ and highly sensitive to experimental noise. Previous work has shown that approximate fitness landscapes with additive and pairwise interactions among residues can accurately describe biophysical properties of proteins, including protein residue-residue contacts^{28, 29}. Additive fitness effects describe the independent contributions of each amino-acid variant to fitness, whereas pairwise epistatic interaction effect describes how pairs of amino acids synergistically (positively) or antagonistically (negatively) impact the fitness^{30, 31}. Here, we construct a statistical model to infer the additive fitness effects and pairwise epistatic interaction that underlie the six local fitness landscapes (see Methods). This model provides an interpretable description of the high-dimensional experimental landscape in the presence of the fitness measurement noise.”

* First sentence of abstract: Is it really correct to say that evolvability “fuels” the evolution? Evolvability is a pre-requisite for the evolution, but selection really fuels it.

Response: Thank you for pointing that out. We agree with the reviewer and have changed the word “fueled” to “enabled” in the first sentence of the abstract.

* Line 45: “Subsequently” doesn’t seem to be the correct word here. Maybe “Consequently” is better?

Response: Thank you also for pointing this out. We have replaced the word “Subsequently” by “Consequently” as suggested.

* Line 102: should be “was” the vaccine strain, not “is” the vaccine strain.

Response: Fixed.

* The methods should better clarify the origin of the non-ectodomain regions of HA and NA. Were these from WSN?

Response: Yes, those are from WSN. We have now clarified in the methods section of the revised manuscript:

“The non-ectodomain region of HA and non-coding region of NA are from A/WSN/33”

* Why are Bk79 and Bei89 so highly correlated? Are these two strains more genetically or structurally similar, particularly in the antigenic region B location in question.

Response: Thank you for the poignant question, which has inspired us to perform an additional analysis. Our additional analysis compared the relationship between the similarity of antigenic site B local fitness landscape between two strains, which is quantified in Fig. 2c, and their pairwise amino-acid sequence identity. This analysis is described in the results section:

“To further understand how the amino-acid sequence in the base of the RBS may influence the local fitness landscape of antigenic site B, we compared the pairwise amino-acid identity in the base of the RBS and the pairwise correlation of the local fitness landscape of antigenic site B among different genetic backgrounds. Here, the base of the RBS is defined as 135, 137, 145, 222, 225, and 226. The amino-acid sequence of the base of the RBS for each of the strains of interest is shown in Supplementary Fig. 8a, and their pairwise amino-acid sequence identity is listed in Supplementary Fig. 8b. The similarity in the local fitness landscape of antigenic site B among different genetic backgrounds, which was quantified in Fig. 2c, correlated well with the pairwise amino-acid sequence identity of the base of the RBS (Fig. 5e, Spearman's rank correlation = 0.66, p-value = 0.007). In contrast, the similarity in the local fitness landscape of antigenic site B among different genetic backgrounds did not have significant correlation with the pairwise amino-acid sequence identity of the entire HA ectodomain (Fig. 5e, Spearman's rank correlation = 0.19, p-value = 0.49). This analysis substantiates the notion that the amino-acid sequence at the base of the RBS can modulate the local fitness landscape of antigenic site B.”

In fact, among all pairwise sequence comparisons of the six strains of interest, Bk79 and Bei89 have the highest sequence identity, both in terms of RBS base and HA ectodomain. This may then be attributable to the high correlation of the antigenic site B local fitness landscapes in Bk79 and Bei89.

AUTHOR IDENTITIES:

We have opted into the options to publish the review and reveal our identities. In accordance with the online reviewer instructions, we are also adding our names here to the end of the review: Allie Greaney and Jesse Bloom.

Reviewer #3 (Remarks to the Author):

In this manuscript, “Major hemagglutinin antigenic site B of human influenza H3N2 viruses has an evolving local fitness landscape.”, Wu and colleagues build off of their previous work analyzing the fitness and mutational landscape of the influenza hemagglutinin, in particular the conserved receptor binding site (RBS) (Wu et al Nat Coms 2018). Specifically, they selected six residues in antigenic site B and performed deep mutational scanning of historical H3N2 viruses. These historical were chosen because they are ~10yrs apart and were also vaccine components. (Unintentionally perhaps, 4 (HK68, BK79, BE89 and BR07) of the 6 strains are also representative strains that define antigenic clusters from Bedford, Smith and colleagues (eLife,2014;

3:e0194)). The replication fitness of 576 variants were tested, and perhaps not surprisingly, the “local” fitness landscape depends on the genetic background (i.e., the specific HA strain). Comparing their previously determined BR07

Minor:

1) Perhaps modify Figure 1 to show the other major antigenic sites on the HA head near site B and RBS. How much overlap (calculated surface area and number of amino acids) is there between these sites?

Response: Thank you for the suggestion. In Figure 1a of the revised manuscript, we have shown all five major antigenic sites (sites A-E) on the HA as suggested. Although these five major antigenic sites are located relatively closely together in the HA globular head domain, they do not overlap. They each contain a unique set of amino acid residues as described in previous studies (Bush et al., 1999; Munoz and Deem, 2005; Skehel et al., 1984; Wiley et al., 1981; Wilson et al., 1981).

2) The authors state that six residues were selected at Antigenic site B because they “reached high occurrence frequency during the natural evolution of H3N2 viruses”; how was this determined / what do the authors mean by this? How do these six residues compare to other residues within antigenic site B in terms of their “occurrence frequency”? Could the authors explain in a little more detail how they arrived at these specific six residues?

Response: In the revised manuscript, we have elaborate how we selected these six residues in the results section:

“Several residues in antigenic site B of influenza HA are also part of the RBS, including residues 156, 158, 159, 190, 193, and 196. Previous studies have shown that mutations at these six residues can affect receptor binding and viral replication fitness^{13, 21, 22, 23}. We compiled an inventory of major amino-acid variants that have reached high occurrence frequency during the natural evolution of human H3N2 viruses over the past 50 years at these six residues of antigenic site B (Fig. 1a-c). This list includes four amino-acid variants at residue 156 (Glu, Lys, Gln, His), four at residue 158 (Gly, Glu, Lys, Asn), three at residue 159 (Ser, Phe, Tyr), two at residue 190 (Asp and Glu), three at residue 193 (Ser, Asn, Phe), and two at residue 196 (Val and Ala).”

3) Were any of the the variants that have reduced fitness tested for affinity for sialic acid? Do the authors feel this would further support their arguments on functional constraint of antigenic site B and the RBS?

Response: Thank you for the comment. In fact, HK68 E190D mutant and Vic11 D190E mutant have previously been tested for receptor binding on glycan array, which is cited as reference #13 (Wu et al., 2018). In the results section of the revised manuscript, we have mentioned the consistency of the results in our previous study with our data presented in the current study:

“For example, this analysis captures the known amino-acid preference at residue 190, where Glu is favorable in early but not recent human H3N2 strains, in terms of both viral replication fitness and receptor binding¹³.”

We actually have an ongoing study that uses glycan arrays to measure the receptor binding of a large panel of HA mutants. This ongoing study is providing a lot of interesting insights into the functional constraint of RBS, including residues in the antigenic site B. We are planning to publish the study independently in the foreseeable future.

4) Supp Fig 7: consider modifying to stereo view for ease of reader interpreting electron density.

Response: As suggested by the reviewer, stereo view is used in Supplementary Fig. 7 of the revised manuscript.

References

Bush, R.M., Bender, C.A., Subbarao, K., Cox, N.J., and Fitch, W.M. (1999). Predicting the evolution of human influenza A. *Science* 286, 1921-1925.

Munoz, E.T., and Deem, M.W. (2005). Epitope analysis for influenza vaccine design. *Vaccine* 23, 1144-1148.

Skehel, J.J., Stevens, D.J., Daniels, R.S., Douglas, A.R., Knossow, M., Wilson, I.A., and Wiley, D.C. (1984). A carbohydrate side chain on hemagglutinins of Hong Kong influenza viruses inhibits recognition by a monoclonal antibody. *Proc Natl Acad Sci U S A* 81, 1779-1783.

Wiley, D.C., Wilson, I.A., and Skehel, J.J. (1981). Structural identification of the antibody-binding sites of Hong Kong influenza haemagglutinin and their involvement in antigenic variation. *Nature* 289, 373-378.

Wilson, I.A., Skehel, J.J., and Wiley, D.C. (1981). Structure of the haemagglutinin membrane glycoprotein of influenza virus at 3 Å resolution. *Nature* 289, 366-373.

Wu, N.C., Thompson, A.J., Xie, J., Lin, C.W., Nycholat, C.M., Zhu, X., Lerner, R.A., Paulson, J.C., and Wilson, I.A. (2018). A complex epistatic network limits the mutational reversibility in the influenza hemagglutinin receptor-binding site. *Nat Commun* 9, 1264.

Reviewers' Comments:

Reviewer #1:

Remarks to the Author:

The authors have satisfactorily addressed all my comments. I have a few further minor comments:

1. Line 96: Were all amino-acid variants at the six residues observed in the natural sequences considered in the analysis? Or some minimum frequency threshold was used to filter them? It would be good to make this clear.
2. Figures 4a and 4b are not referenced in the main text.
3. Lines 294-296: This description of entrenchment and contingency is not required.

Reviewer #2:

Remarks to the Author:

We are recommending the article for acceptance, but still have a few minor suggested revisions:

1. In response to reviewer 1, in line 75 the authors changed "576 variants" to "all possible variants". We actually think the new wording is slightly misleading, because it is not all possible variants at those sites, but just the 576 variants with the amino acids in question. Maybe new wording could clarify this.
2. The fact that wildtype Mos99 was not in the experiment is only mentioned in the Figure 3 legend. This should be mentioned in the main text. In particular, it could be relevant to the discussion on lines 201-208, where it could partly explain the discrepancy between the preferences and KL distance for Mos99 if the RBS sequence at these 6 sites was "rare" and the fitness of the wildtype perhaps due to specific epistatic interactions.
3. The methods say 45,218 HA sequences were downloaded, but lines 708-709 refer to 28,694 HA sequences.

Response to reviewers

Reviewer #1 (Remarks to the Author):

The authors have satisfactorily addressed all my comments. I have a few further minor comments:

1. Line 96: Were all amino-acid variants at the six residues observed in the natural sequences considered in the analysis? Or some minimum frequency threshold was used to filter them? It would be good to make this clear.

Response: This is clarified in the revised manuscript, line 97:

“>90% in any given year”

2. Figures 4a and 4b are not referenced in the main text.

Response: Figures 4a and 4b are now cited in the main text of revised manuscript (lines 229-230).

3. Lines 294-296: This description of entrenchment and contingency is not required.

Response: We have removed the description of entrenchment and contingency in lines 294-296.

Reviewer #2 (Remarks to the Author):

We are recommending the article for acceptance, but still have a few minor suggested revisions:

1. In response to reviewer 1, in line 75 the authors changed "576 variants" to "all possible variants". We actually think the new wording is slightly misleading, because it is not all possible variants at those sites, but just the 576 variants with the amino acids in question. Maybe new wording could clarify this.

Response: We have reworded it as “576 variants of interest” in the revised manuscript.

2. The fact that wildtype Mos99 was not in the experiment is only mentioned in the Figure 3 legend. This should be mentioned in the main text. In particular, it could be relevant to the discussion on lines 201-208, where it could partly explain the discrepancy between the preferences and KL distance for Mos99 if the RBS sequence at these 6 sites was "rare" and the fitness of the wildtype perhaps due to specific epistatic interactions.

Response: In the main text of the revised manuscript (lines 113-116), we have stated that the wild type Mos99 is not include in the deep mutational scanning experiment:

“While the WT sequences of HK68, Bk79, Bei89, Bris07, and NDako16 at the six residues of interest were included in our deep mutational scanning experiment, the WT sequence of Mos99 was not included because it contained a naturally rare variant T196.”

We agree with the reviewer that the discrepancy between the preferences and KL distance for Mos99 may be related to the rare variant carried by Mos99 WT. However, whether such a relationship actually exists is far from clear. Therefore, we try to avoid suggesting such a connection in the manuscript.

3. The methods say 45,218 HA sequences were downloaded, but lines 708-709 refer to 28,694 HA sequences.

Response: Thank you for point out this discrepancy, 45,218 is the correct number. We have fixed the error in lines 725-726 (previous lines 708-709).